# Biogeochemical Layering and Transformation of Particulate Organic Carbon in the Tropical Northwestern Pacific Ocean Inferred from $\delta^{13}C$

## Authors

Detong Tian[1,3], Xuegang Li[1,2,3,4*], Jinming Song[1,2,3,4*], Jun Ma[1,2], Huamao Yuan[1,2,3,4], Liqin Duan[1,2,3,4]

## Affiliations

[1]CAS Key Laboratory of Marine Ecology and Environmental Sciences, Institute of Oceanology, Chinese Academy of Sciences, Qingdao 266000, China

[2]Laboratory for Marine Ecology and Environmental Science, Qingdao Marine Science and Technology Center, Qingdao 266237, China

[3]University of Chinese Academy of Sciences, Beijing 100049, China

[4]Center for Ocean Mega-Science, Chinese Academy of Sciences, Qingdao 266000, China

*Corresponding to*: Xuegang Li (lixuegang@qdio.ac.cn) and Jinming Song (jmsong@qdio.ac.cn).

**Abstract.** Particulate organic carbon (POC) serves as the main carrier of the biological pump and determines its transmission efficiency, yet the transformation processes of POC remain incompletely understood. This study reports the vertical distribution of POC, dissolved inorganic carbon (DIC), $\delta^{13}C$-POC, and $\delta^{13}C$-DIC in the tropical Northwestern Pacific Ocean (TNPO). The research identified three distinct biogeochemical layers governing POC transformation: the POC rapid synthesis-degradation layer (RSDL, 0-300 m), the net degradation layer (NDL, 300-1,000 m), and the stable layer (SL, 1,000-2,000 m). From the top to the bottom of the RSDL, $\delta^{13}C$-POC values decreased by an average of 2.23‰, while the carbon-to-nitrogen ratios (C:N) increased by an average of 2.3:1, indicating the selective degradation of POC. In the NDL, $\delta^{13}C$-POC and $\delta^{13}C$-DIC exhibited a significant negative correlation (r = 0.43, p < 0.05), indicating a net transformation of POC to DIC. In the SL, POC proved to be resistant to degradation, with POC exhibiting the highest C:N (15:1 on average) and the lowest $\delta^{13}C$-POC values (average -27.71‰).

## 1 Introduction

As the most significant carbon reservoir on the earth's surface, the ocean absorbs about 2.6 billion tons of carbon dioxide ($CO_2$) from the atmosphere each year, accounting for 25% of global anthropogenic $CO_2$ emissions (Friedlingstein et al., 2023). After entering the ocean, $CO_2$ initially dissolves in seawater, forming dissolved inorganic carbon (DIC). Subsequently, phytoplankton and photosynthetic bacteria at the ocean surface convert it into organic carbon through photosynthesis. The majority of carbon in the ocean is in the form of DIC, constituting over 98% of the total carbon content, with the remaining 2%

existing as POC and dissolved organic carbon (DOC). Particulate organic carbon (POC) can be

transported to the deep ocean (> 1,000 m) through the biological pump and buried for thousands of years.

This process of carbon sequestration aids in the absorption of $CO_2$ by the ocean, contributing to the

regulation of atmospheric $CO_2$ levels (Longhurst and Glen Harrison, 1989; Turner, 2015). Organic matter

produced in the euphotic layer is the primary food source for heterotrophic communities in the dark

ocean (Smith et al., 2008); once POC is exported from the euphotic layer, microorganisms may rapidly

utilize it, releasing DIC (Song, 2010).

Some studies have shown that unstable components such as proteins and carbohydrates in POC are

preferentially degraded by microorganisms (Eadie and Jeffrey, 1973). However, conducting detailed

quantitative analyses of each POC component in actual investigations is challenging, necessitating the

use of alternative indicators to demonstrate selective degradation. One generally accepted indicator is the

carbon-to-nitrogen ratios (C:N) due to inherent differences in the C:N of various compounds in POC

(Morales et al., 2021). Thus, changes in the C:N during degradation can signify the selective degradation

of POC. Nevertheless, the composition of POC is highly complex, and the C:N of its different

components are not absolute. For example, lipids typically have a higher C:N than proteins, but the

opposite can also occur (Sannigrahi et al., 2005; Hernes and Benner, 2002). Therefore, relying solely on

the C:N to reflect the selective degradation process of POC has significant limitations. The vital activities

of the microbial community in the dark ocean are predominantly driven by heterotrophic respiration

(Herndl et al., 2023), while many autotrophic organisms also use chemical energy to synthesize POC.

There is compelling evidence that chemoautotrophy plays a substantial role in the fixation of DIC in the

(Reinthaler et al., 2010) and the deeper ocean (Passos et al., 2022; Walsh et al., 2009). Consequently,

there is a continuous conversion of POC and DIC throughout the ocean water column. Exploring the

degradation and synthesis of POC in the ocean is imperative to enhance our comprehension of the

biological pump processes.

The DIC in seawater primarily occurs in four chemical forms: $H_2CO_3$, $HCO_3^-$, $CO_3^{2-}$, and $CO_2$. In

comparison, the composition of POC is more complex. POC comprises various organic compounds

originating from living organisms such as phytoplankton, zooplankton, and microorganisms. It also

encompasses fecal particles, cell fragments, and diverse organic substances from external sources. Only

a small fraction of the POC has been accurately identified in terms of molecular structures (Kharbush et

al., 2020). As the depth increases, the readily degradable components in POC are used up, leading to a

more intricate structure of the remaining POC through the transformation process. The remaining refractory POC is even more difficult to identify (Lee et al., 2000). Therefore, it becomes challenging to study the chemical characteristics of POC and its transformation process from itself. The $\delta^{13}C$ value is a crucial indicator that can reveal the origin, migration, and transformation of POC, making it important in the investigation of the marine carbon cycle (Ding et al., 2020; Jeffrey et al., 1983). Compared with POC concentration, $\delta^{13}C$-POC provides a more accurate reflection of the chemical properties of the POC pool and the migration and transformation processes of POC (Close and Henderson, 2020). Similarly, $\delta^{13}C$-DIC can offer insights into important processes within the ocean carbon cycle. As POC settles, it undergoes a series of biogeochemical processes, including synthesis, degradation, and adsorption. Therefore, the isotope fractionation effect in POC is strong, resulting in significant differences in $\delta^{13}C$-POC values at different depths. In contrast, the fractionation of $\delta^{13}C$-DIC is subject to fewer influencing factors, and the DIC concentration in the ocean is notably high, thereby engendering minimal variability in $\delta^{13}C$-DIC values across the ocean water column (Jeffrey et al., 1983). Therefore, $\delta^{13}C$-DIC is more sensitive to the fractionation effect in the ocean carbon cycle. Even slight variations in the $\delta^{13}C$-DIC values can reflect significant processes involved in the migration and transformation of POC (Quay and Stutsman, 2003). Through the analysis of $\delta^{13}C$-POC and $\delta^{13}C$-DIC values, we can enhance our comprehension of the intricate composition, transport, and alteration mechanism of POC, providing us with a more profound insight into the dynamic transformations within the ocean biological pump.

The tropical Northwestern Pacific Ocean (TNPO) is characterized by intricate current patterns and water mass distributions (Hu et al., 2015; Schönau et al., 2022), and it is also known for the highest surface seawater temperatures globally (Jia et al., 2018). High temperatures facilitate the respiration by heterotrophic organisms, promoting the formation of biological hotspots and ultimately enhancing material circulation and energy flow in the upper ocean (0-300 m) (Guo et al., 2023a; Iversen and Ploug, 2013). The air-sea interaction within the TNPO is highly dynamic, exhibiting a shift from being a carbon sink to a carbon source as it extends from higher to lower latitudes (Takahashi et al., 2009; Wu et al., 2005). The complex hydrological characteristics, rapid elemental cycle, and frequent air-sea exchange render the TNPO an ideal laboratory for exploring the ocean carbon cycle. In this research, we collected seawater and particulate matter samples at six stations in the core and boundary regions of the TNPO, and the relationship between DIC, POC, and their stable carbon isotopes was comprehensively analyzed to enhance our understanding of the POC transformation process and the ocean carbon cycle.

## 2 Sampling and Methods

The samples were collected in the TNPO during an expedition on R/V *Kexue* from 16 February to 12 April 2022. A total of 6 stations were set up: EQ-6 (150.99° E, 0.00° N, 1944 m), E142-3 (140.99° E, 12.01° N, 4091 m), E142-7 (140.99° E, 15.99° N, 4725 m), E142-11 (140.99° E, 20.00° N, 462 4m), E142-13 (142.04° E, 0.00° N, 3382 m) and E142-19 (141.99° E, 6.01° N, 2580 m) (Fig. 1). The 12-L Niskin bottles (KC-Denmark, Denmark) mounted on a Conductivity-Temperature-Depth (CTD, Sea-bird SBE911, United States) rosette were used to obtain water samples from the vertical profile of 0-2,000 m at each station for analysis of temperature, salinity, dissolved oxygen (DO), POC, $\delta^{13}$C-POC, particulate nitrogen (PN), DIC, $\delta^{13}$C-DIC, and chlorophyll a (Chl-*a*). The specific sampling and analysis methods are as follows.

**Temperature and salinity:** The temperature and salinity were measured by CTD in situ, with accuracies of $\pm$ 0.001 °C and $\pm$ 0.0003 S m$^{-1}$, respectively (Ma et al., 2024).

**DO:** DO was determined in situ using the manual Winkler titration method, with a measurement precision of 0.22 μmol L$^{-1}$. At each depth, we collected samples in 50 mL brown bottles, added manganese sulfate and alkaline potassium iodide to fix the oxygen, then manually titrated the released iodine with sodium thiosulfate using a calibrated burette to calculate DO concentrations (Bryan et al., 1976; Zuo et al., 2018). The discrete DO samples were used to calibrate the DO concentration data obtained by the CTD sensor.

**POC, $\delta^{13}$C-POC, and PN:** Particle samples were obtained by filtering 2-5 L of seawater onto a GF/F glass filter (0.7 μm, Whatman) that had been combusted in a muffle furnace (450 °C, 4 h) and acid-soaked (0.5 M hydrochloric acid, 24 h). After collection, samples were stored below -20 °C until laboratory analysis. Before analysis, the filter was treated with concentrated hydrochloric acid to remove inorganic carbonates and oven-dried at 60 °C. Afterward, POC, PN concentration, and $\delta^{13}$C-POC value were analyzed using an elemental analyzer and an isotope mass spectrometer (Thermo Fisher Scientific Flash EA 1112 HT-Delta V Advantages, United States) with an accuracy of $\pm$ 0.8‰, $\pm$ 3‰ and $\pm$ 0.2‰, respectively. Blank filters were analyzed alongside samples and exhibited negligible background levels for POC, PN, and $\delta^{13}$C-POC value. Standard reference materials were used to calibrate $\delta^{13}$C and POC, PN measurements, including USGS64 ($\delta^{13}$C = -40.8 $\pm$ 0.04‰, C% = 31.97%, N% = 18.65%, Indiana University), USGS40 ($\delta^{13}$C = -26.39 $\pm$ 0.04‰, C% = 40.8%, N% = 9.52%, Geological Survey, United

States), and Urea #2a ($\delta^{13}C$ = -9.14 ± 0.02‰, C% = 20%, N% = 46.67%, Indiana University). We implemented a quality control protocol by randomly inserting a certified reference material after every 10 samples. The measured values of these reference materials were subsequently plotted against the calibration curve to monitor and verify instrument stability throughout the analytical process (Ma et al., 2021).

**DIC and $\delta^{13}C$-DIC:** Sampling was performed using a 50 ml glass bottle. After the water sample overflowed, 1 ml of the sample was taken out with a pipette and then fixed with saturated mercuric chloride solution to remove the influence of biological activity. After collection, samples were stored in refrigerator at 4 ℃ for later laboratory measurement of DIC concentration using a total DIC analyzer (Apollo SciTech AS-C3, United States) with an accuracy of ± 0.1% (Ma et al., 2020). For calibration, certified reference material (Batch 144, 2031.53 ± 0.62 µmol kg$^{-1}$) provided by the Scripps Institution of Oceanography (University of California, San Diego) was used. $\delta^{13}C$-DIC values automatic analysis was performed using a Thermo Delta-V isotope ratio mass spectrometer (ThermoFisher Scientific MAT 253Plus, United States). For calibration, certified reference materials for $\delta^{13}C$-DIC were used, including GBW04498 ($\delta^{13}C$ = -27.28 ± 0.10‰), GBW04499 ($\delta^{13}C$ = -19.58 ± 0.10‰), and GBW04500 ($\delta^{13}C$ = -4.58 ± 0.12‰), all provided by the Institute of Geophysical and Geochemical Exploration (Chinese Academy of Geological Sciences). We inserted a reference standard every 10 samples, using its measured values to verify instrument stability via the calibration curve.

**Chl-a:** 2 L of water sample after zooplankton removal was filtered onto pre-combusted (450 ℃, 5 h) GF/F filters (0.7 µm, Whatman) and placed in the refrigerator at -20 ℃ before measurement. In the laboratory, the filters were extracted with 90% acetone for 12-24 h, and the concentration was measured using a fluorescence photometer (Turner Designs, United States) For calibration, Chl-a analytical standard (purity ≥ 95.0%) provided by Sigma-Aldrich (SIAL, St. Louis, MO, United States) was used (Ma et al., 2020).

Data analysis was conducted using OriginPro 2021 (v9.8.0.200). Inter-group differences were assessed using t-tests, with statistical significance defined as $p < 0.05$. Linear relationships between variables were examined using least-squares regression, and correlation strength was reported as the Pearson correlation coefficient (r). An $r > 0$ denotes a positive correlation, $r < 0$ a negative correlation, and |r| closer to 1 indicates a stronger linear relationship.

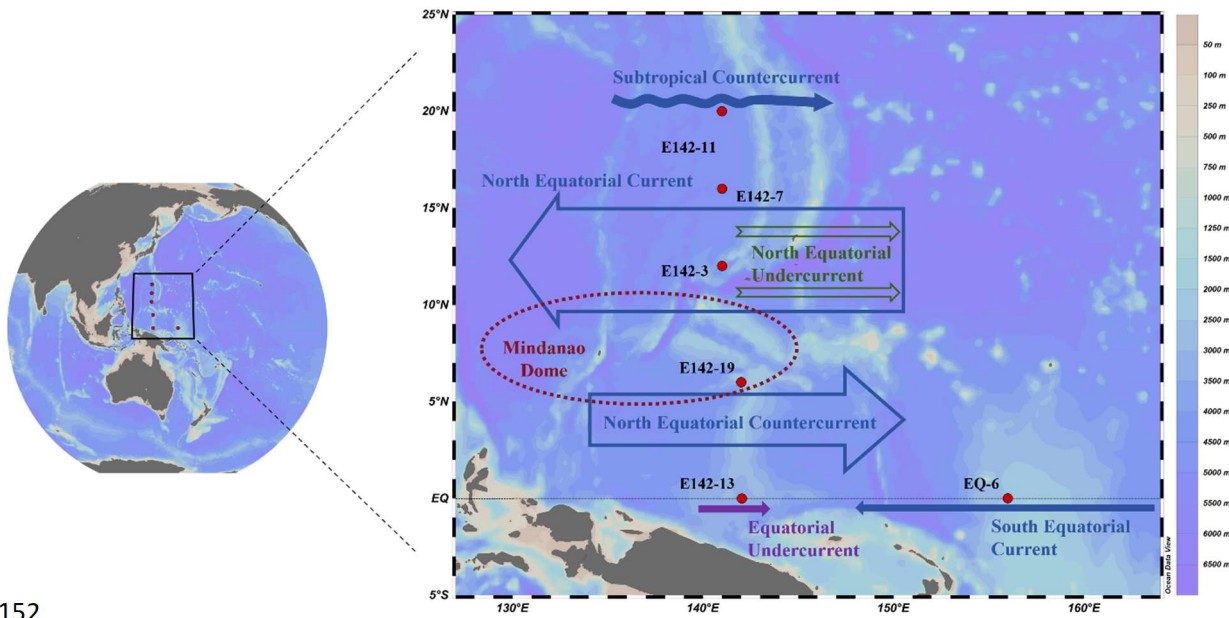


**Figure 1. TPWO sampling stations (red dots in the figure) and ocean current distribution. In the figure, blue**
**represents the ocean currents from the surface to the bottom of the thermocline, mainly Subtropical**
**Countercurrent, North Equatorial Current, North Equatorial Countercurrent, and South Equatorial**
**Current; green represents the ocean currents in the subthermocline, mainly North Equatorial Undercurrent;**
**purple represents the ocean currents from the bottom of the thermocline to the subthermocline, mainly**
**Equatorial Undercurrent.**
**3 Results and Discussion**
**3.1 Hydrological Characteristics**
Except for station E142-11, the remaining five stations are all located at the Western Pacific Warm Pool
(WPWP). The SST of the five stations in the warm pool area was higher, averaging 29.01 ± 0.67 ℃,
while station E142-11 had a lower SST of 25.02 ℃. The strong seawater stratification in the study area
restricted the movement of nutrient-rich water from the deep to the upper ocean, resulting in the region
showing oligotrophic characteristics (Radenac et al., 2013). Therefore, the Chl-*a* concentration at the
deep chlorophyll maximum layer depth (DCMD) was low, with an average of only 0.24 ± 0.04 μg L$^{-1}$.
Based on the fluorescence intensity measured by the CTD in-situ fluorescence sensor, we calculated the
primary production zone depth (PPZD), which is the depth where the fluorescence intensity drops to 10%
of its maximum value above this depth (Owens et al., 2015). Additionally, the mixed layer depth (MLD)
at each station was determined using the temperature threshold method (Table 1) (Thompson, 1976). The
results indicate that the PPZD at each station is deeper than the MLD, suggesting that the POC generated
at these stations does not undergo particularly complex physical mixing after its formation (Buesseler et
al., 2020).

Table 1. Water depth (WD), primary production zone depth (PPZD), mixed layer depth (MLD), deep chlorophyll maximum layer depth (DCMD), and the chlorophyll-a (Chl-*a*) concentration at DCMD for each station.

| Station | Longitude °E | Latitude °N | WD m | PPZD m | MLD m | DCMD m | Chl-*a* µg L$^{-1}$ |
|---------|----------|---------|------|------|-----|------|-------|
| EQ-6 | 155.99 | 0.00 | 1944 | 129 | 65 | 50 | 0.31 |
| E142-3 | 141.00 | 12.01 | 4091 | 216 | 102 | 140 | 0.19 |
| E142-7 | 141.00 | 16.00 | 4725 | 204 | 68 | 150 | 0.25 |
| E142-11 | 140.99 | 20.00 | 4624 | 203 | 42 | 90 | 0.21 |
| E142-13 | 142.04 | 0.00 | 3382 | 165 | 45 | 90 | 0.25 |
| E142-19 | 142.00 | 6.01 | 2580 | 170 | 109 | 100 | 0.21 |

Based on the relationship between potential temperature and salinity (θ-S) (Fig. 2), eight water masses in the study area were identified: North Pacific Tropical Surface Water (NPTSW), North Pacific Subsurface Water (NPSSW), North Pacific Subtropical Mode Water (NPSTMW), North Pacific Intermediate Water (NPIW), North Pacific Deep Water (NPDW), as well as Equatorial Surface Water (ESW), South Pacific Subsurface Water (SPSSW) and South Pacific Intermediate Water (SPIW) (Sun et al., 2008). In the upper ocean, we found that both NPTSSW and SPSSW exhibited high salinity characteristics. The salinity of NPTSSW was distributed between 34.66 and 35.01, while the salinity of SPSSW was distributed between 35.15 and 35.65. In addition, as the water depth increased, the temperature of NPTSSW and SPSSW decreased significantly, with NPTSSW dropping from 27.18 ℃ to 16.21 ℃ and SPSSW dropping from 29.23 ℃ to 14.81 ℃. The representative water mass in the intermediate ocean (300-1000 m) is NPIW, which is characterized by a rapid decrease in temperature (11.44-5.57 ℃) and a slight increase in salinity (~0.3) with increasing water depth. The representative water mass in the deep ocean is NPDW, which has stable properties and slight variations in salinity and temperature. Notably, the water mass distribution at station E142-19 is quite special. Ranging from the subsurface to the deep layer, the water mass properties of this station are relatively stable, showing low-salinity and low-temperature characteristics. This is attributed to the intrusion of both NPIW and SPIW into the station in the intermediate ocean layer. Additionally, the station is situated within the Mindanao Dome upwelling area, where strong upwelling transports low-temperature, low-salinity NPDW from the bottom to the upper layer, enhancing seawater exchange. Consequently, the water at station E142-19 comprises a mixture of diverse water masses.

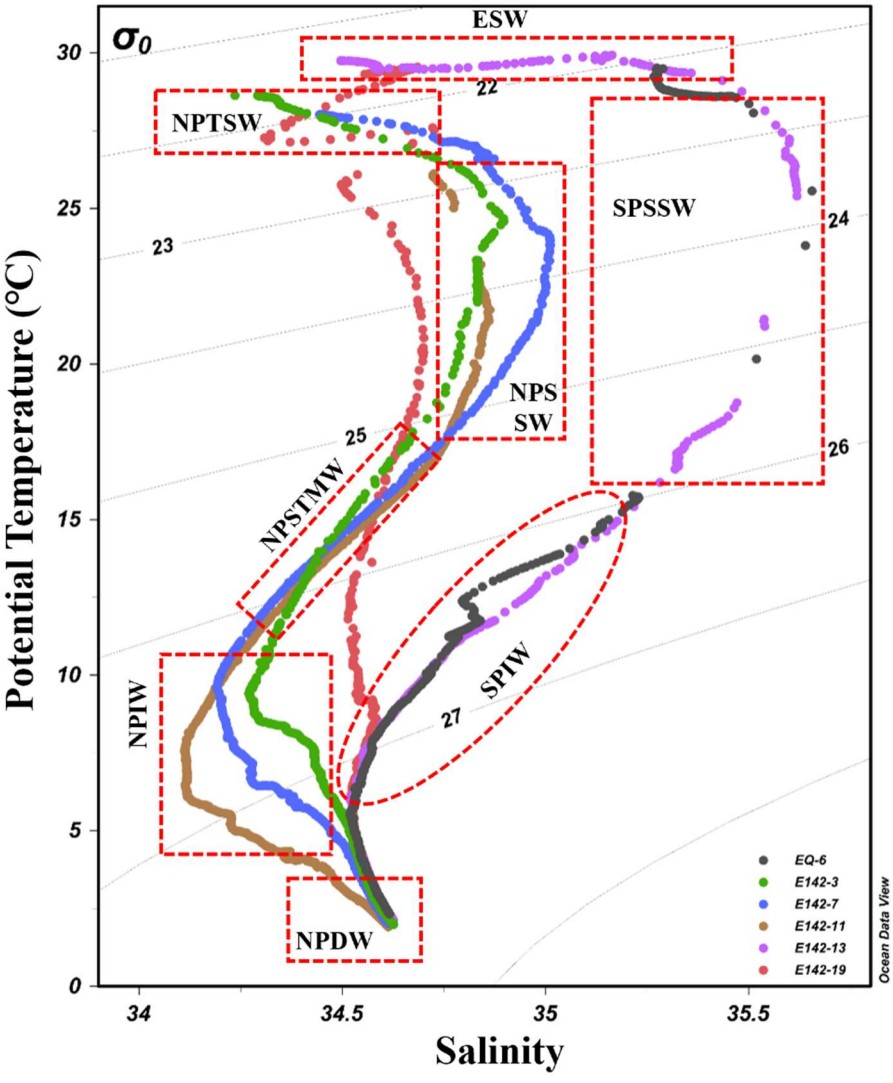

**Figure 2. Relationship between potential temperature (θ) and salinity (S) at each sampling station. The water mass distribution is marked with a dotted line.**

The study area is traversed by six major ocean currents: the South Equatorial Current, the North Equatorial Current, the North Equatorial Undercurrent, the Subtropical Countercurrent, the Equatorial Undercurrent and the North Equatorial Countercurrent (Fig. 1). Among them, the South Equatorial Current flows from east to west along the equator and is characterized by high temperature and low salinity, notably impacting station EQ-6. The North Equatorial Current is a major westward current in the study area, accompanied by a series of eastward undercurrents of the North Equatorial Undercurrent in its lower part; stations E142-3 and E142-7 are mainly affected by them. The Subtropical Countercurrent is characterized by a multi-eddy structure that flows eastward in the subtropical region of the North Pacific and notably impacts station E142-11. The Equatorial Undercurrent is a strong eastward current rich in oxygen and nutrients, which are present in the subsurface layer of the equatorial

Pacific, forming the main body of the thermocline of this area; station E142-13 is deeply affected by it. The North Equatorial Countercurrent is an important current in the tropical Pacific equatorial current system, transporting warm pool water from the western Pacific to the eastern Pacific; Station E142-19 is mainly affected by it. Furthermore, the area features a substantial upwelling system known as the Mindanao Dome, greatly impacting Station E142-19, situated southeast of the Mindanao Dome.

**3.2 Vertical distribution characteristics of POC and $\delta^{13}$C-POC**

The average POC concentration from the surface to the deep chlorophyll maximum layer (DCM, 0-150 m) of the six stations was: E142-19 ($34.12 \pm 3.53$ µg L$^{-1}$) > E142-13 ($31.90 \pm 3.19$ µg L$^{-1}$) > EQ-6 ($31.32 \pm 5.27$ µg L$^{-1}$) > E142-3 ($27.77 \pm 4.78$ µg L$^{-1}$) > E142-7 ($27.43 \pm 1.35$ µg L$^{-1}$) > E142-11 ($26.81 \pm 2.25$ µg L$^{-1}$). The surface POC concentrations at stations E142-13 and EQ-6 were slightly higher than those at other stations, which can be attributed to higher nutrient levels in ESW and SPSSW than in NPTSW and NPTSSW. Notably, the surface POC concentration at station E142-19 was the highest among all stations, primarily due to the intense upwelling associated with the Mindanao Dome that brings nutrient-rich water to the surface, alleviating the nitrogen nutrient limitation of the surface water at this station (Gao et al., 2021).

POC concentrations at all stations demonstrated a decreasing trend with increasing water depth and tended to remain stable in the deep ocean. The most significant drop occurred between the DCM and 600 m (Fig. 3). The seawater within this depth range was abundant in POC and also exhibited relatively high temperature and DO concentration, which likely enhanced the metabolic activities of heterotrophic organisms, thereby accelerating their utilization of POC (Iversen and Ploug, 2013; Sun et al., 2021). The aerobic degradation of POC significantly consumed DO, leading to decreased DO levels and the formation of an oxygen cline (Fig. 3). Since the microbial life activities below the oxygen cline were still active, leading to the continued consumption of DO through POC degradation, the DO cannot be replenished in time. As a result, the low oxygen zone (where DO < 100 µmol L$^{-1}$) exists in the intermediate ocean at all stations (Fig. 3). However, the hypoxic conditions observed at station E142-13 were comparatively less pronounced than those observed at other stations (Fig. 3). This can be attributed to the consistent transport of oxygen and nutrient-rich seawater by the Equatorial Undercurrent to this station, facilitating oxygen replenishment and mitigating deoxygenation (Brandt et al., 2021).

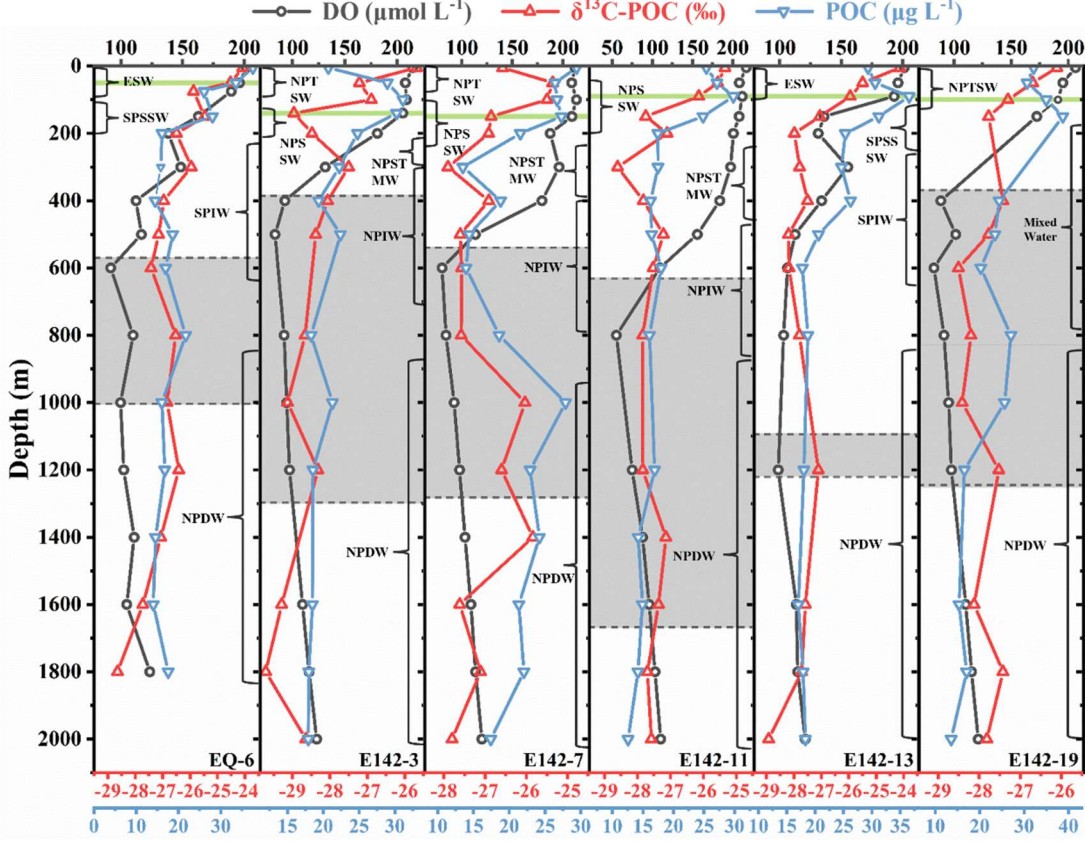

**Figure 3. Vertical distribution of DO concentration, $\delta^{13}$C-POC values, and POC concentration at each sampling station. The gray area marks the hypoxic zone with DO = 100 µmol L$^{-1}$ as the boundary. The green line represents the DCM depth.**

The vertical distribution of $\delta^{13}$C-POC values closely resembles that of POC concentration (Figs. 3, 4a), suggesting that specific $\delta^{13}$C-enriched components may be preferentially degraded during POC degradation. Although the molecular composition of oceanic POC cannot be fully identified, it is generally understood to primarily consist of lipids, amino acids, carbohydrates, nucleic acids, and a small number of heterogeneous components (Kharbush et al., 2020). The metabolic activity of amino acids and carbohydrates is higher than lipids, leading microorganisms to preferentially use these compounds as energy sources, enriching lipids in POC (Hwang et al., 2006; Jeffrey et al., 1983). Previous studies have reported that during the degradation of POC, the carbon isotope fractionation characteristics of amino sugar monomers closely align with changes in $\delta^{13}$C-POC values (Guo et al., 2023b). Moreover, several studies have highlighted that the carbon isotopic composition of lipid monomers does not exhibit significant depletion during POC degradation; in fact, it may even show a trend of enrichment (Close et al., 2014; Häggi et al., 2021). These observations further indicate the preferential degradation of amino acids and carbohydrates in POC. On the other hand, compared with lipids, amino acids and carbohydrates

exhibit higher $\delta^{13}C$ values (Hayes, 1993; Hwang and Druffel, 2003; Schouten et al., 1998). When large
quantities of amino acids and carbohydrates undergo selective degradation, the residual POC will show
low $\delta^{13}C$ value characteristics. Therefore, as POC is continuously consumed in the water column, the
$\delta^{13}C$-POC values will gradually decrease. In addition, lipids have a low nitrogen content in comparison
to amino acids and carbohydrates, leading to a relatively high C:N (Morales et al., 2021). Our findings
demonstrated a strong negative correlation between $\delta^{13}C$-POC values and C:N (Fig. 4b), which implied
that as the water depth increases, $\delta^{13}C$-POC values decreases while the C:N in the remaining POC
increases. This suggests that selective degradation of POC occurs, during which amino acids and
carbohydrates in the POC are preferentially removed, resulting in a relative increase in the proportion of
lipids in the remaining POC (Druffel et al., 2003; Guo et al., 2023a).
However, significant differences were observed among sampling stations in the relationships between
$\delta^{13}C$-POC value and POC concentration, as well as between $\delta^{13}C$-POC value and C:N (Fig. S1). Among
them, the EQ-6 station exhibited the most distinct regression trends. Located within the South Equatorial
Current regime, where the water column is stable and strongly stratified, this station showed strong
correlations ($p < 0.05$) between $\delta^{13}C$-POC value and both POC concentration and C:N, indicating that
degradation-dominated isotopic fractionation processes are prominent in this region (Tuchen et al., 2024).
At station E142-13, under the influence of the Equatorial Undercurrent, continuous nutrient supply and
an active biological pump created a marked gradient in POC content and composition between surface
and subsurface waters (Brandt et al., 2021). As a result, significant correlations ($p < 0.05$) were also
observed between $\delta^{13}C$-POC value and both POC concentration and C:N. In contrast, at station E142-19,
located in the Mindanao Dome upwelling region, the correlation between $\delta^{13}C$-POC value and POC
concentration was not significant ($p > 0.05$). This may be attributed to the upward transport of deep
nutrients and the resuspension or entrainment of aged POC, leading to heterogeneous POC sources and
ages in the water column, which diluted the isotopic fractionation signal associated with degradation
(Gao et al., 2021). Nevertheless, $\delta^{13}C$-POC at this station still exhibited a significant negative correlation
with C:N ($p < 0.05$), indicating the persistence of $\delta^{13}C$ enrichment resulting from organic matter
degradation (Guo et al., 2023b). Moreover, although stations E142-3, E142-7, and E142-11 are located
within the same water mass regime (Fig. 2), their $\delta^{13}C$-POC value correlations with POC concentration
and C:N differed. At station E142-7, no significant correlation was found between $\delta^{13}C$-POC value and
C:N ($p > 0.05$). In contrast, stations E142-3 and E142-11 exhibited significant negative correlations
between δ¹³C-POC value and C:N (p < 0.05), likely due to stepwise degradation processes driven by
water column stratification. These two stations are probably situated at the edges or transition zones of
the water mass, where pronounced stratification limits vertical mixing, thereby creating stronger vertical
gradients in POC degradation and leading to the observed negative correlations (Close et al., 2014; Häggi
et al., 2021). In comparison, station E142-7 may be located near the water mass core, where enhanced
mixing results in a narrower vertical range of C:N (Fig. S1), suggesting a more uniform POC composition
throughout the water column. This homogeneity reduces spatial variability in degradation, thus
weakening the coupling between δ¹³C-POC value and C:N (Meyers, 1997).
In addition, it is noteworthy that in the upper ocean, although there is a significant negative correlation
between δ¹³C-POC values and C:N ratios (p < 0.05), no significant correlation is observed between δ¹³C-
POC values and POC concentration (p > 0.05) (Fig. 4a). This suggests that the fractionation of δ¹³C-POC
at this depth layer is not entirely controlled by selective degradation. Photosynthesis exerts a certain
influence on the fractionation of δ¹³C-POC within this depth range, primarily manifested as an increase
in photosynthetic carbon isotope fractionation with depth, leading to a decrease in δ¹³C-POC values. In
a study conducted in the subtropical North Atlantic, the photosynthetic carbon isotope fractionation
increased by 5.6‰ from the upper to the lower euphotic zone, while the δ¹³C values of the photosynthetic
product, phytol, decreased by 6.3‰ (Henderson et al., 2024). Therefore, although the process of selective
degradation significantly affects the fractionation of δ¹³C-POC, it is still necessary to consider the
regulatory effects of other processes in certain unique marine environments.

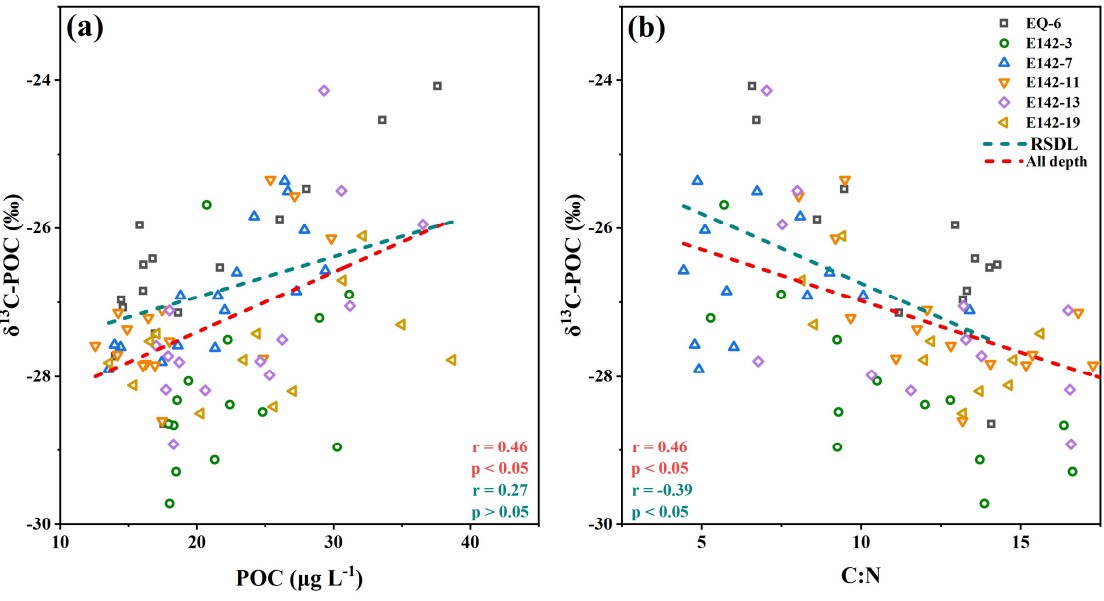

**Figure 4. (a) Relationship between δ¹³C-POC values and POC concentration; (b) Relationship between δ¹³C-**
**POC values and C:N. Red and green lines indicate regressions for the full water column and the rapid**
**synthesis-degradation layer, respectively.**

### 3.3 Vertical distribution characteristics of DIC and $\delta^{13}$C-DIC

Among the six stations, only the equatorial stations E143-13 and EQ-6 exhibited average upper DIC
concentrations exceeding 2000 μmol kg$^{-1}$, with values of 2036 and 2054 μmol kg$^{-1}$, respectively. This
phenomenon can be attributed to the fact that the surface water masses at these stations are composed of
high-temperature and high-salinity ESW (Fig. 2). Although high temperatures generally hinder the
dissolution of $CO_2$, they can accelerate the rate of $CO_2$ release by heterotrophic organisms. Meanwhile,
high salinity increases the ionic strength and buffering capacity of seawater, promoting DIC
accumulation (Zeebe and Wolf-Gladrow, 2001). These factors collectively contribute to the high DIC
concentrations observed in the surface layers of these two stations. The average upper DIC concentration
at station E142-19 was the next highest, reaching 1992 μmol kg$^{-1}$. This is due to upwelling at this station,
which transports deep, high-DIC seawater to the intermediate ocean. Consequently, this station also
recorded the highest average intermediate DIC concentration among the six stations, at 2184 μmol kg$^{-1}$.
Furthermore, since stations E142-3, E142-7, and E142-11 are predominantly influenced by the same
water mass across all depths, their DIC concentrations are relatively similar at each depth (Fig. 5). The
average DIC concentrations of all six stations in the upper ocean, intermediate ocean, and deep ocean
were 2004 ± 65, 2147 ± 35, and 2234 ± 26 μmol kg$^{-1}$, respectively. There was a significant increase in
DIC concentration from the upper to the deep ocean (Fig. 5). In the upper ocean, DIC concentrations are
lower due to photosynthetic uptake, whereas the decomposition of POC at intermediate depths releases
inorganic carbon, causing elevated DIC levels with depth. In the deep ocean, a small amount of POC
may still degrade, and, along with the release of DIC driven by decreasing carbonate saturation,
contributes to a gradual further increase in DIC concentrations.
Moreover, we observed surface $\delta^{13}$C-DIC values ranging from -0.55 to 0.45‰ (average 0.12‰) in the
research region, which is significantly lower than those reported in studies conducted in the Pacific region
in the 1990s (Quay et al., 2017; Quay and Stutsman, 2003). This suggests that the ocean has absorbed
more anthropogenic $CO_2$ over the years. The surface $\delta^{13}$C-DIC value of station E142-11 was the lowest
among the six stations, only -0.55‰, while the surface $\delta^{13}$C-DIC value of station EQ-6 was the highest,
reaching 0.45‰. This is because station E142-11 was located at the strongest atmospheric $CO_2$ net sink
area, while station EQ-6 was located at the atmospheric $CO_2$ net source area (Zhong et al., 2022). The
sea-air exchange at station E142-11 was sufficient, leading to a lower $\delta^{13}C$-DIC value in its surface water,
as it was more likely to reach isotopic equilibrium with atmospheric $CO_2$. In contrast, the surface water
of station EQ-6 was more susceptible to seawater mixing and biological primary production influences.
The higher $\delta^{13}C$-DIC values observed in the surface water of station EQ-6 can be attributed to the isotope
fractionation caused by the consumption of a substantial amount of $CO_2$ by biological primary production
(Quay et al., 2003). In analyzing the vertical distribution of $\delta^{13}C$-DIC, the findings revealed a pronounced
decrease in $\delta^{13}C$-DIC values at each station (Fig. 5), consistent with the $\delta^{13}C$-POC variations observed
in the upper ocean (Fig. 6d). Within this depth range, the average decrease in $\delta^{13}C$-POC values was
2.23‰, while the average decrease of $\delta^{13}C$-DIC values was 0.30‰, with $\delta^{13}C$-DIC reaching its minimum
value in the subsurface. However, in the intermediate ocean layer, unlike $\delta^{13}C$-POC, $\delta^{13}C$-DIC values
increased first and then stabilized (Fig. 5). Therefore, distinct differences exist in the overall change
trends of $\delta^{13}C$-DIC values and $\delta^{13}C$-POC values in the ocean water column. Since the mutual conversion
between POC and DIC was ongoing, this conversion process will inevitably cause changes in $\delta^{13}C$-POC
and $\delta^{13}C$-DIC. Generally, the variation range of $\delta^{13}C$-POC values was more significant than that of $\delta^{13}C$-
DIC values, indicating the more complex biogeochemical processes experienced by POC (Meyer et al.,
2016; Schmittner et al., 2013). This difference is also partly due to the much larger size of the DIC pool
compared to the POC pool (Jeffrey et al., 1983). The high DIC concentration in the ocean buffers its
isotopic variability, resulting in minimal changes in $\delta^{13}C$-DIC values across the water column, whereas
the smaller POC pool is more sensitive to localized biogeochemical processes, leading to greater
variability in $\delta^{13}C$-POC values.

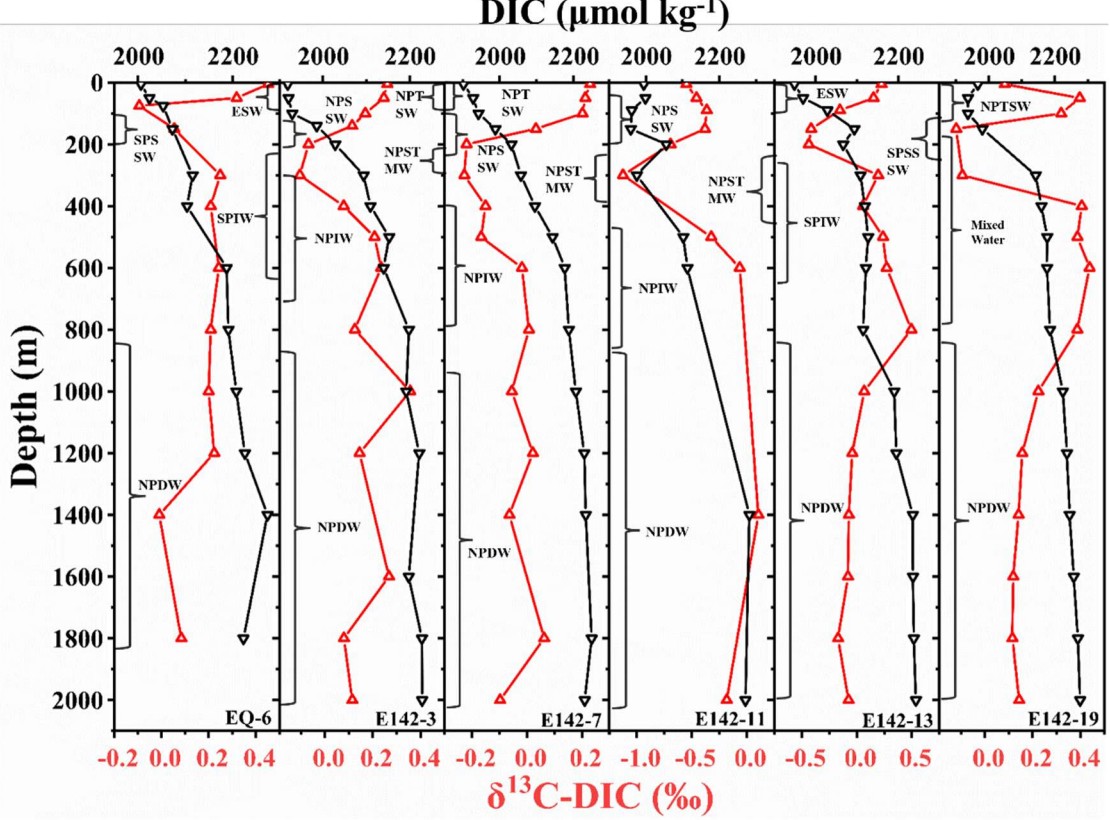

**356**

**357** Figure 5. Vertical distribution of DIC concentration and δ¹³C-DIC values at each sampling station. The black

**358** line represents DIC, and the red line represents δ¹³C-DIC.

**359** **3.4 Transformation characteristics of POC in different water layers**

**360** According to the distribution characteristics of δ¹³C-POC and δ¹³C-DIC values, we divided the ocean

**361** water column into three biogeochemical layers: the POC rapid synthesis-degradation layer (RSDL, 0-

**362** 300 m), the net degradation layer (NDL, 300-1,000 m) and the stable layer (SL, 1,000-2,000 m). Within

**363** the RSDL, POC undergoes concurrent synthesis and degradation (Calbet and Landry, 2004). The

**364** synthesis of POC likely exceeded its degradation from the surface to the DCM layer, while the

**365** degradation of POC appeared to dominate below the DCM. In this layer, the δ¹³C-POC values and C:N

**366** exhibited a pronounced negative correlation, while no significant correlation is observed between δ¹³C-

**367** POC values and POC concentration ($p > 0.05$) (Figs. 4a, 6a). Therefore, the decrease of δ¹³C-POC values

**368** in this layer was dominated by the selective degradation of POC and by photosynthesis. Both δ¹³C-POC

**369** and δ¹³C-DIC values decrease with increasing depth in the RSDL (Figs. 3, 5), and they exhibit a

**370** significant positive correlation within this layer (Fig. 6d). Although the degradation of POC typically

**371** lowers the δ¹³C value of DIC, as the δ¹³C value of POC is lower than that of DIC, the significant decline

**372** in δ¹³C-DIC values observed in the RSDL, when considering the substantial difference in magnitude

between the POC pool and the DIC pool, suggests the influence of additional processes. Specifically, the phytoplankton and photosynthetic bacteria in the upper ocean tend to use the light $^{12}CO_2$ in the seawater for photosynthesis; thus the $\delta^{13}$C-DIC values of the near surface ocean at all stations were relatively high. However, light intensity diminishes with increasing depth, which is unfavorable for photosynthesis. This leads to the accumulation of $^{12}CO_2$ produced by the respiration of heterotrophic communities. Consequently, the $\delta^{13}$C-DIC values in this layer steadily declined with depth (Ge et al., 2022). In the NDL, sunlight was extremely weak, and photosynthesis was nearly absent. Heterotrophic communities dominate, leading to a continuous decrease in POC concentration and a corresponding increase in DIC concentration (Figs. 3, 5). Generally, the degradation of POC would be expected to lower the $\delta^{13}$C value of DIC. However, in this layer, $\delta^{13}$C-POC values showed a significant negative correlation with both C:N and $\delta^{13}$C-DIC values (Fig. 6b, e), indicating the influence of additional processes on $\delta^{13}$C-DIC fractionation. The NDL often encompasses low-oxygen zones (Fig. 3), which are known to favor the activity of chemoautotrophic microorganisms. Compared to aerobic environments, the energy required for microorganisms to fix inorganic carbon into organic carbon is lower under low-oxygen conditions (Hugler and Sievert, 2011; Mccollom and Amend, 2005). During this process, chemoautotrophic microorganisms preferentially utilize lighter $^{12}$C isotopes, leading to the enrichment of $\delta^{13}$C in the remaining DIC pool. This microbial activity may explain the observed increase in $\delta^{13}$C-DIC values in the NDL. In the SL, the POC concentration remained consistently low. $\delta^{13}$C-POC values did not correlate significantly with either C:N or $\delta^{13}$C-DIC (Fig. 6c, f). This was because the easily degradable components in POC had been completely consumed in the RSDL and NDL, and the remaining components were relatively refractory. As a result, the conversion of POC to DIC was rare in SL, leading to an absence of a clear link between $\delta^{13}$C-POC and $\delta^{13}$C-DIC.

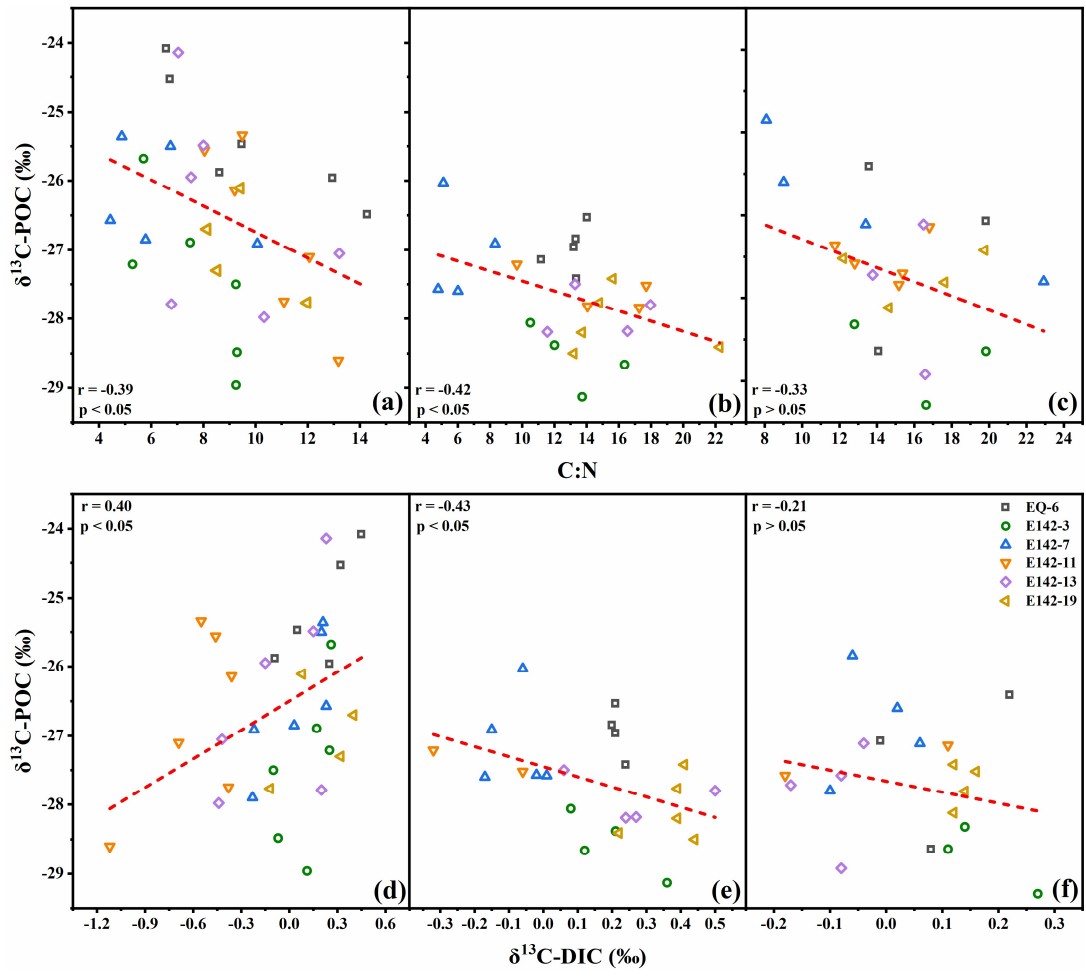

**Figure 6. Relationships between δ¹³C-POC and C:N at different depths: (a) 0-300 m, (b) 300-1,000 m, (c) 1,000-2,000 m, and between δ¹³C-POC and δ¹³C-DIC at different depths: (d) 0-300 m, (e) 300-1,000 m, (f) 1,000-2,000 m.**

## 4 Conclusions

In general, this study investigated the transformation characteristics of POC in the tropical northwest Pacific Ocean based on the $\delta^{13}$C perspective. Our findings revealed three distinct stages of POC behavior in the ocean: rapid synthesis-degradation, net degradation, and stable existence. Below the RSDL, the selective degradation of POC dominated the changes in $\delta^{13}$C-POC. The C:N ratio data in RSDL and NDL indicate an increase in the proportion of refractory lipids in POC, relative to more labile components such as amino acids and carbohydrates. Consequently, in the SL, POC was found to be stable with a slow degradation rate. The fractionation of $\delta^{13}$C-DIC in the ocean is influenced by both the production and degradation processes of POC. Within the RSDL, $\delta^{13}$C-DIC fractionation is predominantly governed by

primary production, whereas within the NDL and SL, it is primarily influenced by the degradation
process of POC.
Although we utilized $\delta^{13}C$-POC and $\delta^{13}C$-DIC to assess the overall transformation characteristics of POC,
the specific synthesis and decomposition ratios of POC are still challenging to determine. Further
research is needed on the monomer carbon isotopic composition of POC (lipids, amino acids, etc.) to
enhance our understanding of the transformation process of POC.
**Data Availability.** The data files used in this paper are available at (Tian et al., 2024).
**Competing interest.** The authors declare that they have no conflict of interest.
**Author contribution.** Detong Tian: Investigation, Data Curation, Writing-original draft. Xuegang Li
and Jinming Song: Conceptualization, Funding acquisition, Writing-review & editing. Jun Ma, Funding
acquisition. Huamao Yuan, Liqin Duan: Writing - Review & Editing.
**Acknowledgments.** This work was supported by the National Key Research and Development Program
(grant no. 2022YFC3104305), National Natural Science Foundation of China (grant nos.42176200,
42206135); Laoshan Laboratory (grant nos. LSKJ202204001, LSKJ202205001). We appreciate the
crews of the R/V *Kexue* for sampling assistance during the cruise of NORC2021-09 supported by the
National Natural Science Foundation of China (project no. 42049909).

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
