# Peer review of "Biogeochemical Layering and Transformation of"

_EGUsphere, 2024_

## Author Comment (AC1)

**Response to RC1**

**Title: "Biogeochemical Layering and Transformation of Particulate Organic Carbon in the Tropical Northwestern Pacific Ocean Inferred from $\delta^{13}C$"**

**In this document, we present the response to Referee's comments repeated in blue.**

*Thank you very much for taking the time to review our manuscript and providing insightful and constructive comments. Your feedback has been invaluable in helping us identify areas for improvement and enhancing the overall quality of our work. We have carefully considered each of your suggestions and made the necessary revisions. Below, we provide detailed responses to each of your comments:*

**1. While this paper represents new data on POM elemental and stable isotope composition (which is always welcome) and the interpretation given appears fairly senseful, though largely speculative. But because the data set is limited to 'classical' parameters of POM and does not provide more specific data about POM composition (e.g. isotopic composition of specific components) the paper falls short in substantially improving our insights in the fate of POM in the oceanic water column with regard to existing literature.**

*Thank you for highlighting both the strengths and limitations of our study. To address your concern regarding the speculative nature of our interpretation, we have incorporated additional references in the discussion section to provide a more robust context. Specifically, we have included studies on the carbon isotopic fractionation of amino sugar monomers and lipid monomers during POC degradation (revised manuscript L224-230).*

*"Previous studies have reported that during the degradation of POC, the carbon isotope fractionation characteristics of amino sugar monomers closely align with changes in $\delta^{13}C$-POC (Guo et al., 2023b). Moreover, several studies have highlighted that the carbon isotopic composition of lipid monomers does not exhibit significant depletion during POC degradation; in fact, it may even show a trend of enrichment (Close et al., 2014; Häggi et al., 2021). These observations further indicate the preferential degradation of amino acids and carbohydrates in POC."*

*We have also emphasized in the conclusion section the need for future studies to focus on the carbon isotopic composition of specific POC components (e.g., lipids and amino acids) to deepen our understanding of POC transformation processes (revised manuscript L324-326).*

**References:**

Guo, J., et al. (2023b). Stable carbon isotopic composition of amino sugars in heterotrophic bacteria and phytoplankton: Implications for assessment of marine organic matter degradation. Limnology and Oceanography, 68, 2814–2825. http://doi.org/10.1002/lno.12468

Close, H. G., et al. (2014). Lipid and $^{13}$C signatures of submicron and suspended particulate organic matter in the Eastern Tropical North Pacific: Implications for the contribution of Bacteria. Deep Sea Research Part I, 85, 15–34. http://doi.org/10.1016/j.dsr.2013.11.005

Häggi, C., et al. (2021). Impact of selective degradation on molecular isotope compositions in oxic and anoxic marine sediments. Organic Geochemistry, 153. http://doi.org/10.1016/j.orggeochem.2021.104192

**2. A further major shortcoming of this paper is the one-dimensional (surface to deep) approach used when interpreting the data, despite the apparent complexity of ocean currents and counter currents in the studied area. No use is made of T-S, nutrient data to inform on mixed layer depth, DCM position and to identify major water masses and possible impacts of advection processes on observed profiles.**

We appreciate your comments regarding the oversimplified approach in our interpretation. To address this issue, we have added a potential temperature-salinity (T-S) diagram (Figure 1, revised manuscript Figure 2) and detailed the identification of water masses in the study area (revised manuscript L150–170).

*"Based on the relationship between potential temperature and salinity (θ-S) (Fig. 2), eight water masses in the study area were identified: North Pacific Tropical Surface Water (NPTSW), North Pacific Subsurface Water (NPSSW), North Pacific Subtropical Mode Water (NPSTMW), North Pacific Intermediate Water (NPIW), North Pacific Deep Water (NPDW), as well as Equatorial Surface Water (ESW), South Pacific Subsurface Water (SPSSW) and South Pacific Intermediate Water (SPIW). In the upper ocean (0-300 m), we found that both NPTSSW and SPSSW exhibited high salinity characteristics. The salinity of NPTSSW was distributed between 34.66 and 35.01, while the salinity of SPSSW was distributed between 35.15 and 35.65. In addition, as the water depth increased, the temperature of NPTSSW and SPSSW decreased significantly, with NPTSSW dropping from 27.18℃ to 16.21℃ and SPSSW dropping from 29.23℃ to 14.81℃. The representative water mass in the middle ocean (300-1000 m) is NPIW, which is characterized by a rapid decrease in temperature (11.44-5.57℃) and a slight increase in salinity (~0.3) with increasing water depth. The representative water mass in the deep ocean (1000-2000 m) is NPDW, which has stable properties and slight changes in salinity and temperature. Notably, the water mass distribution at station E142-19 is quite special. Ranging from the subsurface to the deep layer, the water mass properties of this station are relatively stable, showing low-salinity and low-temperature characteristics. This is attributed to the intrusion of both North Pacific Intermediate Water (NPIW) and South Pacific Intermediate Water (SPIW) into the station in the mid-ocean region. Additionally, the station is situated within the MD upwelling area, where strong upwelling transports low-temperature, low-salinity North Pacific Deep Water (NPDW) from the bottom to the upper layer, enhancing seawater*

*exchange. Consequently, the water at station E142-19 comprises a mixture of diverse water masses."*

[Figure]

**Figure 1. Relationship between potential temperature (θ) and salinity (S) at each sampling station. The data points at each station are marked with hollow circles of different colors. (Source: Tian et al. (2025), manuscript under review)**

We also revised the discussion section to highlight the influence of water mass nutrient conditions on POC concentrations (revised manuscript L193-194):

*"Since the nutrient concentration in ESW and SPSSW is higher than that in NPTSW and NPTSSW, the surface POC concentrations at stations E142-13 and EQ-6 were slightly higher than those at other stations."*

Additionally, we updated the Vertical distribution of DO, $\delta^{13}$C-POC, and POC (Figure 2, revised manuscript Figure 3) and marked the positions of the DCM on the diagrams for clarity.

[Figure]

**Figure 2. Vertical distribution of DO, δ¹³C-POC, and POC concentration at each sampling station. The gray area marks the hypoxic zone with DO = 100 μmol/L as the boundary. The green line represents the DCM depth.**

**3. The method section should be more detailed, since no information on sample preservation, standards, references used, corrections applied .. is given.**

We appreciate your suggestion to provide more detailed information in the methods section. In response, we have extensively revised this section to include details on sample preservation, the use of standard reference materials, and the applied corrections (revised manuscript L105–135):

*"DO: Water samples were collected, fixed, and titrated according to the classic Winkler method, the precision of which was 2.2×10-3 μmol/L (Bryan et al., 1976; Zuo et al., 2018). The discrete DO samples were used to calibrate the DO concentration data obtained by the CTD sensor.*

*POC, δ¹³C-POC, and PN: Particle samples were obtained by filtering 2-5 L of seawater onto a GF/F glass filter (0.7 μm, Whatman) that had been combusted in a muffle furnace (450°C, 4 h) and acid-soaked (0.5 M hydrochloric acid (HCl), 24 h). The filter was treated with HCl to remove inorganic carbonates and oven-dried at 60°C. After collection, samples were stored below -20 °C until laboratory analysis. Afterward, POC, PN concentration, and δ¹³C-POC were analyzed using an elemental analyzer and an isotope mass spectrometer (Thermo Fisher Scientific Flash EA 1112 HT-Delta V Advantages, United States) with an accuracy of ± 0.8‰ and ± 0.2‰, respectively. Standard reference materials were used to calibrate δ13C and POC, PN measurements, including USGS64 (δ¹³C = -40.8 ± 0.04‰, C% = 31.97%, N% = 18.65%,*

*Indiana University), USGS40 ($\delta^{13}C$ = -26.39 ± 0.04‰, C% = 40.8%, N% = 9.52%, Geological Survey, United States), and Urea #2a ($\delta^{13}C$ = -9.14 ± 0.02‰, C% = 220%, N% = 46.67%, Indiana University) (Ma et al., 2021).*

*DIC and $\delta^{13}C$-DIC: Sampling was performed using a 50 ml glass bottle. After the water sample overflowed, 1 ml of the sample was taken out with a pipette and then fixed with saturated mercuric chloride solution to remove the influence of biological activity. After collection, samples were stored in refrigerator at 4°C for later laboratory measurement of DIC concentration using a total DIC analyzer (Apollo SciTech AS-C3, United States) with an accuracy of ± 0.1% (Ma et al., 2020). For calibration, certified reference material (Batch 144, 2031.53 ± 0.62 μmol/kg) provided by the Scripps Institution of Oceanography (University of California, San Diego) was used. $\delta^{13}C$-DIC automatic analysis was performed using a Thermo Delta-V isotope ratio mass spectrometer (ThermoFisher Scientific MAT 253Plus, United States). For calibration, certified reference materials for dissolved inorganic carbon ($\delta^{13}C$-DIC) were used, including GBW04498 ($\delta^{13}C$ = -27.28 ± 0.10‰), GBW04499 ($\delta^{13}C$ = -19.58 ± 0.10‰), and GBW04500 ($\delta^{13}C$ = -4.58 ± 0.12‰), all provided by the Institute of Geophysical and Geochemical Exploration (Chinese Academy of Geological Sciences).*

*Chl -a: 2 L of water sample after zooplankton removal was filtered onto pre-combusted (450°C for 5 h) GF/F filters (0.7 μm, Whatman) and placed in the refrigerator at −20°C before measurement. In the laboratory, the filters were extracted with 90% propanol for 12-24 h, and the concentration was measured using a fluorescence photometer (Turner Designs, United States) (Ma et al., 2020)."*

**We believe these revisions provide a clearer and more comprehensive description of our methods, addressing both your concerns and the needs of readers.**

**Once again, we thank you for your valuable feedback and constructive suggestions, which have significantly improved the quality of our manuscript. We hope the revisions we have made address your concerns and enhance the scientific rigor of the study.**

---

## Author Comment (AC2)

**Response to RC2**

**Title: "Biogeochemical Layering and Transformation of Particulate Organic Carbon in the Tropical Northwestern Pacific Ocean Inferred from $\delta^{13}C$"**

In this document, we present the response to Referee's comments repeated in blue.

Thank you very much for your constructive feedback and valuable suggestions.

1. I appreciate the fact that a T-S diagram has been provided. The depth scale in color gradient is not really necessary as it renders the reading of the graph more difficult and since in any case isopycnals are shown. So, either remove the depth scale or consider plotting another parameter instead of depth (DIC, d13C, or may be try plotting nitrate or phosphate, for which you have data).

We appreciate your suggestion regarding the depth scale in the T-S diagram. In response, we have revised the diagram by removing the depth scale (Fig. 1). We believe this enhances the clarity of the water mass distribution.

[Figure]

**Figure 1. Relationship between potential temperature (θ) and salinity (S) at each sampling station. The water mass distribution is marked with a dotted line. (Source: Tian et al. (2025), manuscript under review)**

**2. You mention NPSSW, which does not appear in the T-S plot; do you mean NTPSSW?**

**We apologize for the error in the T-S plot, where NTPSSW was incorrectly labeled. We have corrected this to NPSSW, as originally intended (Fig. 1).**

**3. In order to address my concern about absence of discussion how water masses might possibly affect the vertical distributions of studied parameters, I wonder whether you could indicate the position of the different water masses present at each of the stations (vertical profiles in Figs 2 and 4).**

**Thank you for your valuable feedback. In response, we have updated Figures 2 (Fig. 2) and 4 (Fig. 3) to include the vertical distribution of different water masses at each station. This addition aims to better illustrate the influence of water masses on the vertical profiles of the parameters under investigation. We hope these revisions adequately address your concern.**

[Figure]

**Figure 2. Vertical distribution of DO, δ¹³C-POC, and POC concentration at each sampling station. The gray area marks the hypoxic zone with DO = 100 μmol/L as the boundary. The green line represents the DCM depth.**

[Figure]

**Figure 3. Vertical distribution of DIC concentration and δ¹³C-DIC at each sampling station. The black line represents DIC, and the red line represents δ¹³C-DIC.**

**4. In Fig. 1 you show an arrow marked 'NGCUC' along the PNG coast. It was not identified and discussed.**

We appreciate your attention to detail regarding the 'NGCUC' in Figure 1. We have removed the label to avoid confusion (Fig. 4).

[Figure]

**Figure 4. TPWO sampling stations (red dots in the figure) and ocean current distribution. In the figure, blue represents the ocean currents from the surface to the bottom of the thermocline, mainly STCC, NEC, NECC, and SEC; green represents the ocean currents in the subthermocline, mainly NEUC; purple represents the ocean currents from the bottom of the thermocline to the subthermocline, mainly EUC.**

We hope these revisions effectively address your concerns. Once again, we are grateful for your thoughtful comments.

---

## Author Response (AR2)

**Response to RC4**

**Title: "Biogeochemical Layering and Transformation of Particulate Organic Carbon in the Tropical Northwestern Pacific Ocean Inferred from $\delta^{13}$C"**

**In this document, we present the response to Referee's comments repeated in blue.**

We sincerely appreciate the thorough review of our revised manuscript by you and the reviewers. We have carefully addressed all the remaining comments and further refined the manuscript, with particular attention to enhancing the discussion of inter-station variability and clarifying the description of process rates. Below we provide point-by-point responses to your specific comments along with detailed explanations of the corresponding revisions.

**1. This is a study with valuable data of POC, DIC and their isotopes for an important region around the equator. This is a review of the revised version of the manuscript. I think most of the comments by the two referees of the first version of the manuscript were satisfactorily accounted for. The vertical profiles of POC and DIC are now more connected to the horizontal currents, which was a major critique. What could be more emphasized and discussed is the changes between the different stations. In their analysis, often all stations are taken together, for example in Figures 4 and 6. There might be interesting differences between the stations, as they are also influenced by different water masses.**

We thank you for this insightful suggestion. In the revised manuscript, we have now expanded the discussion of inter-station variability in Section 3.2 and provided a new figure (Figure 1/Revised Manuscript Figure S1) to highlight differences between $\delta^{13}$C-POC, POC concentration, and C:N ratios across stations (Revised Manuscript Line 265-292): "*However, significant differences were observed among sampling stations in the relationships between $\delta^{13}$C-POC value and POC concentration, as well as between $\delta^{13}$C-POC value and C:N (Fig. S1). Among them, the EQ-6 station exhibited the most distinct regression trends. Located within the South Equatorial Current regime, where the water column is stable and strongly stratified, this station showed strong correlations (p < 0.05) between $\delta^{13}$C-POC value and both POC concentration and C:N, indicating that degradation-dominated isotopic fractionation processes are prominent in this region (Tuchen et al., 2024). At station E142-13, under the influence of the Equatorial Undercurrent,*

*continuous nutrient supply and an active biological pump created a marked gradient in POC content and composition between surface and subsurface waters (Brandt et al., 2021). As a result, significant correlations ($p < 0.05$) were also observed between $\delta^{13}$C-POC value and both POC concentration and C:N. In contrast, at station E142-19, located in the Mindanao Dome upwelling region, the correlation between $\delta^{13}$C-POC value and POC concentration was not significant ($p > 0.05$). This may be attributed to the upward transport of deep nutrients and the resuspension or entrainment of aged POC, leading to heterogeneous POC sources and ages in the water column, which diluted the isotopic fractionation signal associated with degradation (Gao et al., 2021). Nevertheless, $\delta^{13}$C-POC at this station still exhibited a significant negative correlation with C:N ($p < 0.05$), indicating the persistence of $\delta^{13}$C enrichment resulting from organic matter degradation (Guo et al., 2023b). Moreover, although stations E142-3, E142-7, and E142-11 are located within the same water mass regime (Fig. 2), their $\delta^{13}$C-POC value correlations with POC concentration and C:N differed. At station E142-7, no significant correlation was found between $\delta^{13}$C-POC value and C:N ($p > 0.05$). In contrast, stations E142-3 and E142-11 exhibited significant negative correlations between $\delta^{13}$C-POC value and C:N ($p < 0.05$), likely due to stepwise degradation processes driven by water column stratification. These two stations are probably situated at the edges or transition zones of the water mass, where pronounced stratification limits vertical mixing, thereby creating stronger vertical gradients in POC degradation and leading to the observed negative correlations (Close et al., 2014; Häggi et al., 2021). In comparison, station E142-7 may be located near the water mass core, where enhanced mixing results in a narrower vertical range of C:N (Fig. S1), suggesting a more uniform POC composition throughout the water column. This homogeneity reduces spatial variability in degradation, thus weakening the coupling between $\delta^{13}$C-POC value and C:N (Meyers, 1997)."*

[Figure]

**Figure 1. /Revised Manuscript Figure S1. Relationships between $\delta^{13}$C-POC and (a) POC concentration and (b) C:N across sampling stations.**

References

Tuchen, F. P., Perez, R. C., Foltz, G. R., McPhaden, M. J., and Lumpkin, R.: Strengthening of the Equatorial Pacific Upper-Ocean Circulation Over the Past Three Decades, J. Geophys. Res. Oceans, 129, http://doi.org/10.1029/2024jc021343, 2024.

Brandt, P., Hahn, J., Schmidtko, S., Tuchen, F. P., Kopte, R., Kiko, R., Bourlès, B., Czeschel, R., and Dengler, M.: Atlantic Equatorial Undercurrent intensification counteracts warming-induced deoxygenation, Nat. Geosci., 14, 278-282, http://doi.org/10.1038/s41561-021-00716-

1, 2021.

Gao, W., Wang, Z., Li, X., and Huang, H.: The increased storage of suspended particulate matter in the upper water of the tropical Western Pacific during the 2015/2016 super El Niño event, J. Oceanol. Limnol., 39, 1675-1689, http://doi.org/10.1007/s00343-021-0362-0, 2021.

Guo, J., Achterberg, E. P., Shen, Y., Yuan, H., Song, J., Liu, J., Li, X., and Duan, L.: Stable carbon isotopic composition of amino sugars in heterotrophic bacteria and phytoplankton: Implications for assessment of marine organic matter degradation, Limnology and Oceanography, 68, 2814-2825, http://doi.org/10.1002/lno.12468, 2023b.

Close, H. G., Wakeham, S. G., and Pearson, A.: Lipid and $^{13}$C signatures of submicron and suspended particulate organic matter in the Eastern Tropical North Pacific: Implications for the contribution of Bacteria, Deep Sea Res. Part I Oceanogr. Res. Pap., 85, 15-34, http://doi.org/10.1016/j.dsr.2013.11.005, 2014.

Häggi, C., Pätzold, J., Bouillon, S., and Schefuß, E.: Impact of selective degradation on molecular isotope compositions in oxic and anoxic marine sediments, Org. Geochem., 153, http://doi.org/10.1016/j.orggeochem.2021.104192, 2021.

Meyers, P. A.: Organic geochemical proxies of paleoceanographic, paleolimnologic, and paleoclimatic processes, Org. Geochem., 27, 213-250, http://doi.org/10.1016/s0146-6380(97)00049-1, 1997.

**2. Something that was not always corrected as a response to the previous review is the mentioning of rates of change by the authors. The present study did not measure any rates, but still at some places in the manuscript rates are suggested. I have the impression that in some cases the authors mean the general processes that occur and are well-known. In that case they should not use the past tense as this would suggest that they measured it. If it is general knowledge, one should use the present tense. I have touched upon that in the below comments.**

Thank you for pointing this out. To avoid confusion,

① We have removed all implicit rate suggestions (e.g., "rapid") unless citing published rate measurements.

② Clearly distinguished between our observations and established knowledge (now using present tense for the latter).

**3. There are too many abbreviations of currents, regions, etc. in the text. Please use less of those for enhancing the readability. Only generally accepted and well-known abbreviations such as SST, POC, CTD are useful. Even when not using those abbreviations increases the length of the text, this is not an issue in online publishing.**

We appreciate this suggestion. To improve readability, we revised the manuscript to reduce the use of abbreviations, especially for regional currents and zones. Full names are now provided in the main text and figure/table captions. Furthermore, to maintain the clarity and visual appeal of the figures, we have opted to retain the abbreviations for the relevant water masses.

**List of comments:**

**1. L34 Define POC here, as it is used for the first time. Even when it has been defined in the abstract, this must be done in the main text, because abstract and main text are considered separate.**

Thank you for catching this oversight. We have now explicitly defined particulate organic carbon (POC) at its first mention in the main text for clarity.

**2. L34 delete: Despite being in minimal quantities (Because this has no relation with the following part of the sentence)**

We agree this phrase was unnecessary and have deleted it to improve the flow of the sentence.

**3. L38 in instead of: from**

We appreciate this correction. The text now reads (Revised Manuscript Line 37-38):: "*Organic matter produced in the euphotic layer...*"

**4. L39 add "may": … microorganisms MAY rapidly utilize it …**

Thank you for this suggestion. We have added "*may*" to reflect that rapid utilization of POC by microorganisms is a potential process rather than a measured result.

**5. L50-52 I think the beginning of the sentence with "Although" is confusing. Both parts of the sentence are equivalent.**

This is a helpful point. The sentence has been restructured to clarify the parallelism (Revised Manuscript Line 50-52): "*The vital activities of the microbial community in the dark ocean are predominantly driven by heterotrophic respiration (Herndl et al., 2023), while many autotrophic organisms also use chemical energy to synthesize POC.*"

**6. L53 change to: There is compelling evidence that …**

We have adopted this more precise phrasing (Revised Manuscript Line 53): "*There is compelling evidence that …*"

**7. L54 oxygen minimum zone (as the abbreviation already indicates)**

Apologies for the typographical error in the terminology, which has been corrected to "*oxygen minimum zone*" Additionally, since this term appears fewer than three times in the text, the abbreviation "OMZ" has been removed to enhance readability.

**8. L58 delete primarily (there are only those four forms)**

Thank the you for noting this redundancy. The word "primarily" has been removed.

**9. L66 $\delta^{13}$C value**

Thank you. To ensure clarity, "$\delta^{13}$C" has been changed to "$\delta^{13}$C value".

**10. L67 delete significantly**

We agree this modifier was unnecessary and have deleted it.

**11. L94 delete at the end of the sentence: process**

Thank you for the suggestion. We have removed "process" to improve conciseness.

**12. L96 Please be more specific about the dates of the expedition. Is there a cruise report that can be cited?**

We have added the exact dates (Revised Manuscript Line 95-96): "*The samples were collected in the TNPO during an expedition on R/V Kexue from 16 February to 12 April 2022.*" We regret that the cruise report is not publicly available, so we did not cite it.

**13. L104 Please give precision and/or accuracy of temperature and salinity**

Thank you for the suggestion. We have supplemented the relevant information in the Methods section (Revised Manuscript Line 104-105): "*The temperature and salinity were measured by CTD in situ, with accuracies of ± 0.001 °C and ± 0.0003 S/m, respectively (Ma et al., 2024).*"
References
Ma, J., Wen, L., Li, X., Dai, J., Song, J., Wang, Q., Xu, K., Yuan, H., and Duan, L.: Different fates of particulate matters driven by marine hypoxia: A case study of oxygen minimum zone in the Western Pacific, Mar. Environ. Res., 200, 106648, http://doi.org/10.1016/j.marenvres.2024.106648, 2024.

**14. L106-107 The precision for the DO determination is given as 0.0022 μmol/L. I do not know any method of DO determination with such a precision. This must be a mistake. Please explain and correct.**

We sincerely apologize for this error. The text now reads: (Revised Manuscript Line 106-107): "*DO was determined in situ using the manual Winkler titration method, with a measurement precision of 0.22 μmol/L.*"

**15. L106 and further: Is this an automated Winkler method or titration by hand?**

We employed the manual Winkler titration method for DO determination, and the corresponding text has been revised accordingly (Revised Manuscript Line 106-110): "*DO was determined in situ using the manual Winkler titration method, with a measurement precision of 0.22 μmol/L. At each depth, we collected samples in 50 mL brown bottles, added manganese sulfate and alkaline potassium iodide to fix the oxygen, then manually titrated the released iodine with sodium thiosulfate using a calibrated burette to calculate DO concentrations (Bryan et al., 1976; Zuo et al., 2018).*"

**16. L115-117 Three variables are given at the beginning of the sentence, but at the end only two accuracies are given. Please correct**

We apologize for the omission. The sentence now states (Revised Manuscript Line 116-118): "*POC, PN concentration, and $\delta^{13}$C-POC value were analyzed using an elemental analyzer and an isotope mass spectrometer (Thermo Fisher Scientific Flash EA 1112 HT-Delta V Advantages, United States) with an accuracy of ± 0.8‰, ± 3‰ and ± 0.2‰*"

**17. L11 and further: In the response to reviewers the authors described how blank corrections were treated. As these blanks may play a large role, in particular at low concentrations for**

**example for samples below the euphotic zone, this is information that should also occur here in the methods section. Please add this.**

We appreciate this suggestion and have added the following statement (Revised Manuscript Line 119-120): "*Blank filters were analyzed alongside samples and exhibited negligible background levels for POC, PN, and $\delta^{13}$C-POC value.*"

**18. L145-148 Please give the full names of the currents as shown in the figure in the caption. All these different current names are confusing to the reader.**

We agree this improves clarity. The caption now lists (Revised Manuscript Line 153-158): "*…blue represents the ocean currents from the surface to the bottom of the thermocline, mainly Subtropical Countercurrent, North Equatorial Current, North Equatorial Countercurrent, and South Equatorial Current; green represents the ocean currents in the subthermocline, mainly North Equatorial Undercurrent; purple represents the ocean currents from the bottom of the thermocline to the subthermocline, mainly Equatorial Undercurrent.*"

**19. L164 Please add the full names of the abbreviations used in the table in the caption**

Thank you for your suggestion. We have expanded the caption (Revised Manuscript Line 174-176): "*Table 1. Water depth (WD), primary production zone depth (PPZD), mixed layer depth (MLD), deep chlorophyll maximum layer depth (DCMD), and the chlorophyll-a (Chl-a) concentration at DCMD for each station.*"

**20. L166 As to the water masses of the region, please add a reference.**

We appreciate the suggestion. A supporting reference has been added:

*Sun, C., Xu, J., Liu, Z., Tong, M., and Zhu, B.: Application of Argo Data in the Analysis of Water Masses in the Northwest Pacific Ocean, Marine Science Bulletin, 10, 2008*

**21. L175 "The representative water mass in the middle ocean" change to: "The representative water mass at intermediate water depths …"**

We sincerely appreciate this suggestion for improved precision in terminology. To maintain terminological consistency with subsequent sections, we have revised "*middle ocean*" to "i*ntermediate ocean*" throughout the manuscript.

**22. L178 variations instead of changes**

Thank you for this suggestion. "*Changes*" has been replaced with "*variations*" to better describe the context.

**23. Figure 2: In the response to reviewers, in the caption of this figure, Tian et al 2025 manuscript under review is shown. Please discuss in the main text what the theme of that manuscript is and where it differs from the present manuscript.**

We greatly appreciate the opportunity to clarify this point. After careful consideration, we have removed the mention of Tian et al. (2025) from the figure caption to maintain focus on the current study. That work examines TEP (transparent exopolymer particles) biogeochemistry in the same region but uses different datasets and has distinct objectives. Additionally, the manuscript under

review will cite the present study instead.

**24. L190-192 Please refer to Fig. 1 here.**

Thank you for catching this oversight. We have added the appropriate reference to Figure 1 in this section.

**25. L218 leads instead of: led**

We appreciate this correction regarding verb tense. The text now uses present tense ("leads") to properly describe this general biogeochemical relationship.

**26. L219 and further: Which change and acceleration do you mean? You did not measure changes, right? Please reformulate if you mean something different.**

We sincerely appreciate this important critique. We have carefully revised this section to remove any implication that we measured rates of change. The text now reads (Revised Manuscript Line 229-231): "***The aerobic degradation of POC significantly consumed DO, leading to decreased DO levels and the formation of an oxygen cline (Fig. 3).***"

**27. L222 cannot instead of: could not**

Thank you for this suggestion. "***Could not***" has been replaced with "***cannot***" for grammatical precision.

**28. L223 exists instead of: emerges**

We appreciate your careful attention to precise language. We have revised "***emerges***" to "***exists***" to enhance clarity.

**29. L232 Referring to Fig. 2 is not correct here and Fig. 3 does not have 3a.**

We apologize for this error and thank you for identifying it. The reference has been corrected to (Figs. 3, 4a).

**30. Figure 4: At the bottom left a correlation coefficient is given and a probability. In both panels p < 0.001. However, at the bottom right another p is shown, bigger or smaller than 0.05. Please explain how the statistics work here and how the significance was calculated.**

We are grateful for this suggestion to improve methodological transparency. We have added this explanation to the Methods section (Revised Manuscript Line 147-151): "***Data analysis was conducted using OriginPro 2021 (v9.8.0.200). Inter-group differences were assessed using t-tests, with statistical significance defined as $p < 0.05$. Linear relationships between variables were examined using least-squares regression, and correlation strength was reported as the Pearson correlation coefficient (r). An $r > 0$ denotes a positive correlation, $r < 0$ a negative correlation, and |r| closer to 1 indicates a stronger linear relationship.***" Additionally, we have redrawn Figure 4 to improve its clarity and readability (Fig. 2/ Revised Manuscript Fig. 4).

[Figure]

**Figure 2. / Revised Manuscript Figure 4. a.** Relationship between $\delta^{13}$C-POC values and POC concentration; **b.** Relationship between $\delta^{13}$C-POC values and C:N. Red and green lines indicate regressions for the full water column and the rapid synthesis-degradation layer, respectively.

**31. L279 intermediate instead of middle, L280 intermediate -depth instead of: mid-layer, L283 again: intermediat"**

Thank you for this suggestion. We have systematically replaced all instances of "***middle ocean***" and "***mid-layer***" with "***intermediate …***" throughout the manuscript to maintain terminological precision.

**34. L285-286 "Affected by photosynthesis, DIC increases gradually in the upper ocean." This is not correct. Photosynthesis utilizes $CO_2$ and thus reduce DIC in the surface layer. In the deeper layers the DIC values are thus higher. Please change to wording so that more detail for this contention is conveyed.**

We sincerely appreciate this important correction. The revised text now accurately states (Revised Manuscript Line 324-325): "***In the upper ocean, DIC concentrations are lower due to photosynthetic uptake…***"

**35. L286 intermediate depths**

    **L286, 287 delete rapid, unless you give a reference that this process is rapid**

    **L286-287 Change to: … the decomposition of POC releases inorganic carbon, causing elevated DIC throughout the intermediate water column …**

    **L288 Change to: … a small amount of POC may still degrade …**

We are grateful for this opportunity to clarify. The text now reads (Revised Manuscript Line 325-328): "***…the decomposition of POC at intermediate depths releases inorganic carbon, causing elevated DIC levels with depth. In the deep ocean, a small amount of POC may still degrade, and, along with the release of DIC driven by decreasing carbonate saturation, contributes to a gradual further increase in DIC concentrations.***"

**36. L293 delete: as atmospheric CO₂ concentrations have increased**

We agree this phrase was unnecessary. "" has been removed to improve conciseness.

**37. L295 delete: among the six stations**

Thank you for this suggestion. "" has been deleted to avoid redundancy.

**38. L303, 304, 306 A rapid decrease and a rapid decline is mentioned. What decreases and decline is meant here (how were these determined) and how can you know it is rapid without having measured the rates?**

We appreciate this important critique. "*Rapid*" has been replaced with "*pronounced*" to describe the observed trend without implying a quantified rate (Revised Manuscript Line 341-345): "*In analyzing the vertical distribution of $\delta^{13}$C-DIC, the findings revealed a pronounced decrease in $\delta^{13}$C-DIC values at each station (Fig. 5), consistent with the $\delta^{13}$C-POC variations observed in the upper ocean (Fig. 6d). Within this depth range, the average decrease in $\delta^{13}$C-POC values was 2.23‰, while the average decrease of $\delta^{13}$C-DIC values was 0.30‰, with $\delta^{13}$C-DIC reaching its minimum value in the subsurface.*"

**39. L305 Figures 4 and 5d are referred to. Is that correct? There are no subpanels a,b,c,d in Figure 5.**

We apologize for this error. The text now correctly references Figure 5 and Figure 6d (Revised Manuscript Line 341-343): "*In analyzing the vertical distribution of $\delta^{13}$C-DIC, the findings revealed a pronounced decrease in $\delta^{13}$C-DIC values at each station (Fig. 5), consistent with the $\delta^{13}$C-POC variations observed in the upper ocean (Figs. 6d).*"

**40. L326 "POC was rapidly degraded while being synthesized" How do you know? You did not measure that, did you? Is this a contention based on the literature? If yes, then the present tense should be used.**

Thank you for this suggestion. A literature citation (Calbet & Landry, 2004) has been added to support the statement. The sentence was revised to the present tense (Revised Manuscript Line 362-363): "*Within the RSDL, POC undergoes concurrent synthesis and degradation (Calbet and Landry, 2004).*"

Calbet, A. and Landry, M. R.: Phytoplankton growth, microzooplankton grazing, and carbon cycling in marine systems, Limnology and Oceanography, 49, 51-57, http://doi.org/10.4319/lo.2004.49.1.0051, 2004.

**41. L330 Again rapid decrease. How was that determined?**

We have deleted "" and revised to (Revised Manuscript Line 367-368): "*Therefore, the decrease of $\delta^{13}$C-POC values in this layer was dominated by…*"

**42. L331 … and by photosynthesis (add: by, because as it is now it may be ambiguous)**

Thank you for this helpful comment. We have revised the sentence to (Revised Manuscript Line 367-368) "*…the decrease of δ13C-POC values in this layer was dominated by the selective*

*degradation of POC and by photosynthesis." to clarify the relationship and avoid ambiguity.*

**43. L337 tend instead of tended**

Thanks for pointing out the tense issue. We have revised it as suggested.

**44. L338 … of the near surface ocean … (because the layer concerned is more than the just the surface)**

We appreciate this suggestion for precision. We have clarified the sentence by using "***…of the near surface ocean…***" to reflect the accurate depth range.

**45. L341 I think you mean declined with depth, right? If yes, then this should be mentioned.**

Thank you for this suggestion. The sentence has been revised to (Revised Manuscript Line 378): "***Consequently, the $\delta^{13}$C-DIC values in this layer steadily declined with depth (Ge et al., 2022).***"

**46. L352 … activity may explain the observed … (add: may, because you did not measure it)**

Thank you for your suggestion. We have added this important qualifier (Revised Manuscript Line 389): "***This microbial activity may explain the observed increase***…"

**47. Figure 6: How was the significance of the relationships determined? Please explain.**

We appreciate this comment and have addressed it by adding detailed descriptions of the statistical methods, as outlined in our response to Comment #30.

**48. L530 Please delete strange symbols**

We apologize for this formatting issue. The strange symbols in references have been verified and corrected. (Revised Manuscript Line 570-572): "***Schmittner, A., Gruber, N., Mix, A. C., Key, R. M., Tagliabue, A., and Westberry, T. K.: Biology and air-sea gas exchange controls on the distribution of carbon isotope ratios ($\delta^{13}$C) in the ocean, Biogeosciences, 10, 5793-5816, http://doi.org/10.5194/bg-10-5793-2013, 2013.***"

We are truly grateful for the valuable suggestions provided during this final pre-acceptance stage, which have helped strengthen the rigor and clarity of our conclusions. We have meticulously addressed all remaining issues and ensured the revised version meets the journal's standards. Should any additional minor adjustments be required, we stand ready to make them promptly. We look forward to the final publication of our work.

---

## Author Response (AR3)

**Response to RC4**

**Title: "Biogeochemical Layering and Transformation of Particulate Organic Carbon in the Tropical Northwestern Pacific Ocean Inferred from $\delta^{13}C$"**

**In this document, we present the response to Referee's comments repeated in blue.**

Thank you for your careful review and helpful feedback. We have addressed your two final comments as follows:.

**1. L53-54 You forgot to insert Oxygen Minimum Zone before (Reinthaler et al., 2010)**

We apologize for this oversight. We have now inserted "Oxygen Minimum Zone" before the citation (Reinthaler et al., 2010) as suggested.

**2. L104 Why did you delete the brand of the CTD? I think Seabird should still be mentioned here.**

Thank you for pointing out this issue. We have reinstated the brand name ("Sea-bird SBE911, United States") for the CTD to maintain consistency and clarity.

In addition. We have revised the Author Contribution section to use author initials as requested.

The manuscript has been updated accordingly, and we will ensure these changes are reflected in the final submission. Please let us know if any further adjustments are needed.